# Daptomycin versus Vancomycin for the Treatment of Methicillin-Resistant *Staphylococcus aureus* Bloodstream Infection with or without Endocarditis: A Systematic Review and Meta-Analysis

**DOI:** 10.3390/antibiotics10081014

**Published:** 2021-08-21

**Authors:** Alberto Enrico Maraolo, Agnese Giaccone, Ivan Gentile, Annalisa Saracino, Davide Fiore Bavaro

**Affiliations:** 1First Division of Infectious Diseases, Cotugno Hospital, AORN Dei Colli, 80131 Naples, Italy; 2Section of Infectious Diseases, Department of Clinical Medicine and Surgery, University of Naples Federico II, 80131 Naples, Italy; agnesegiaccone94@gmail.com (A.G.); ivan.gentile@unina.it (I.G.); 3Department of Biomedical Sciences and Human Oncology, Clinic of Infectious Diseases, University of Bari, 70121 Bari, Italy; annalisa.saracino@uniba.it (A.S.); davidebavaro@gmail.com (D.F.B.)

**Keywords:** daptomycin, vancomycin, MRSA, *Staphylococcus aureus*, bloodstream infection, endocarditis

## Abstract

Background: Methicillin-resistant Staphylococcus aureus (MRSA) is an important cause of invasive infections, mainly bloodstream infections (BSI) with or without endocarditis. The purpose of this meta-analysis was to compare vancomycin, the mainstay treatment, with daptomycin as therapeutic options in this context. Materials: PubMed, Embase and the Cochrane Database were searched from their inception to 15 February 2020. The primary outcome was all-cause mortality. Secondary outcomes included clinical failure, infection recurrence, persistence of infection, length-of-stay, antibiotic discontinuation due to adverse events (AEs) and 30-day re-admission. This study was registered with PROSPERO, CRD42020169413. Results: Eight studies (1226 patients, 554 vs. 672 in daptomycin vs. vancomycin, respectively) were included. No significant difference in terms of overall mortality was observed [odds ratio (OR) 0.73, 95% confidence interval (CI) 0.40–1.33, I^2^ = 67%]. Daptomycin was associated with a significantly reduced risk of clinical failure (OR 0.58, 95% CI 0.38–0.89, I^2^ = 60%), as confirmed by pooling adjusted effect sizes (adjusted OR against the use of vancomycin 1.94, 95%CI 1.33–1.82, I^2^ = 41%), and was linked with fewer treatment-limiting AEs (OR 0.15, 95%CI 0.06–0.36, I^2^ = 19%). No difference emerged between the two treatments as secondary outcomes. Results were not robust to unmeasured confounding (E-value lower than 95% CI 1.00 for all-cause mortality). Conclusions: Against MRSA BSI, with or without endocarditis, daptomycin seems to be associated with a lower risk of clinical failure and treatment-limiting AEs compared with vancomycin. Further studies are needed to better characterize the differences between the two drugs.

## 1. Introduction

Methicillin-resistant Staphylococcus aureus (MRSA) bacteremia and endocarditis are still associated with a significant disease burden, as well as with a remarkable mortality, ranging from 20% to at least 40% [1,2,3,4], which is generally higher than the death rate linked with methicillin-susceptible S. aureus (MSSA) infections, considering that methicillin-resistance is an independent risk factor for mortality [5]. Several strategies to optimize the management of S. aureus bloodstream infections (BSI) with or without endocarditis have been suggested, including the development of a series of bundles (follow up blood cultures, early source control, echocardiography, etc.) [6] in order to standardize the “gold standard” of care.

Indeed, as demonstrated by the existing literature, adherence to evidence-based procedures in all cases of S. aureus BSI could significantly reduce mortality, duration of hospitalization, costs, complications and, possibly, duration of treatment [7,8].

The most important lingering question involves the best therapeutic choice for MRSA BSI. To date, authoritative guidelines back the use of vancomycin (VAN) or daptomycin (DAP), alternatively [9]. Although VAN is still considered the “backbone” of MRSA infections, serious concerns have been raised over the years, stemming from factors such as slow bactericidal activity, low therapeutic index, limited tissue penetration, unfavorable safety profile, as well as increasing reports of resistance and failure [10]. In this regard, the principal alternative for the treatment of MRSA BSI and endocarditis is DAP, a lipopeptide featuring significant bactericidal activity and an appreciable tolerability profile [9]. DAP was approved for BSI and right-sided endocarditis by S. aureus following the results of a non-inferiority trial by Fowler and colleagues [11].

To date, several studies investigated and compared the efficacy and safety of both drugs in the context of MRSA BSI and endocarditis; however, no definitive agreement has been reached and no meta-analytic comparison has been made. Consequently, the purpose of this meta-analysis is to determine which agent is more effective at reducing mortality and achieving clinical cure. Additionally, the study evaluated whether there is any difference related to infection recurrence, persistence of infection, hospital length-of-stay and adverse events occurrence.

## 2. Materials and Methods

### 2.1. Study Design

The study methodology was consistent with the recommendations provided by the Preferred Reporting Items for Systemic Reviews and Meta-analyses (PRISMA) statement [12], and reporting was conducted according to the newly updated version of the statement (PRISMA 2020) [13]. The corresponding checklist is provided in Appendix A.

The study protocol was submitted and registered with the PROSPERO International Prospective Register of Systematic Reviews (study ID: CRD42020169413) before the start of the literature search in April 2020, and was updated in January 2021.

The research question was developed in according with the PICO framework [14]: patients with BSI/endocarditis by MRSA (population); DAP as prescribed drug (intervention); VAN as control being the reference drug (comparison); mortality as primary endpoint, along with other secondary ones (outcome).

### 2.2. Eligibility Criteria

All randomized controlled trials utilizing a parallel study design, or observational studies, including cohort and case–control studies comparing the use of daptomycin and vancomycin for the treatment of MRSA BSI/endocarditis, were deemed eligible for inclusion.

However, studies not reporting mortality as an outcome, or animal and in vitro studies, along with those involving MRSA BSI-related pneumonia or those including less than 10 subjects per arm, were excluded.

Finally, conference abstracts, commentaries, editorials and review papers were not included in the analysis.

### 2.3. Search Strategy

A thorough search through PubMed/Medline, Embase and the Cochrane Library databases was performed by using appropriate words combinations: daptomycin, methicillin-resistant Staphylococcus aureus (or MRSA), bloodstream infection (or BSI) and endocarditis. Studies in humans, published from the databases’ inception to 15 February 2020, were included. In addition, a hand-search of reference lists of all relevant articles was performed. Neither geographical nor language restriction was applied. The detailed search strategy is described in Appendix A.

### 2.4. Data Extraction

Two investigators (AEM, DFB) independently screened each article for eligibility and inclusion using an electronic spreadsheet (Microsoft Corp., Redmond, WA, USA) in a two-step process. In the first step, screening by title and abstract was carried out to identify studies potentially fulfilling the established criteria for inclusion. The full texts of the articles entering the second step were retrieved for careful assessment in order to pinpoint all eligible studies for final inclusion. The reference lists of these articles were also screened to identify additional pertinent studies. Any discrepancy was solved by consensus among the entire study group.

Data from retrieved papers were extracted and inserted into an Excel spreadsheet as follows: authors, country, publication year, type of study, number of patients, baseline features of the population under investigation (such as mean/median age, main comorbidities, Charlson Score, APACHE Score, Pitt Bacteremia Score), follow-up timing, drug doses and durations, companion drugs, covariates (in case of multivariate analysis), outcome measures.

### 2.5. Outcomes Assessed

Our meta-analysis question was to compare efficacy and safety of daptomycin vs. vancomycin in the setting of methicillin-resistant Staphylococcus aureus (MRSA) bloodstream infection (BSI) and endocarditis.

The primary outcome was all-cause mortality. Secondary outcomes included clinical failure, infection recurrence, persistence of infection, length-of-stay, safety profile (antibiotic discontinuation due to adverse events [AEs]) and 30-day re-admission. Definitions used in the primary studies for clinical failure, infection recurrence and persistence of infection were adopted and explicitly stated.

### 2.6. Quality Assessment

Two reviewers (AEM, DFB) independently assessed study quality of selected papers according to pre-specified tools, and any disagreement was solved by general consensus. Protocol observational studies (case–control and cohort) were appraised through an adapted version of the Newcastle–Ottawa scale (NOS) [15]. In this scale, observational studies were scored across three domains by resorting to the following parameters: selection (four questions), comparability (two questions) and ascertainment of the outcome of interest (three questions). Studies with a total score of 7 or more were considered high-quality, studies with a score between 4 and 6 were deemed moderate-quality and low-quality studies had a score of less than 4.

The Cochrane risk of bias tool for clinical trials, in its updated version (RoB2), was intended to be used for randomized trials [16].

The Grading of Recommendations Assessment, Development and Evaluation (GRADE) tool was employed for the overall assessment of the body of evidence in the systematic review in order to gauge the certainty (quality) of the findings [17].

### 2.7. Statistical Analyses

For every study included, the estimate was represented by odds ratio for binary outcomes and mean difference for continuous outcomes calculated with their 95% confidence interval (CI). Meta-analyses were conducted through a random-effects model (DerSimonian and Laird) in order to yield a pooled effect size [18]. If not reported, means and standard deviations for continuous outcomes were derived from sample size, median, IQR and minimum and maximum values, as described by Wan and colleagues [19]. Heterogeneity between studies was assessed by I2 index (values of 25%, 50%, and 75% indicating low, moderate, high heterogeneity) [20]. Anticipating a limited number of retrieved studies, Doi plot and the Luis Furuya–Kanamori (LFK) index were used to determine the presence of small-study effects instead of funnel plot with Egger’s regression test [21], in order to identify asymmetry as an indicator of publication or other biases. A symmetrical mountain-like graph with values of the LFK index within ±1 suggests no asymmetry; between ±1 and ±2 indicates minor asymmetry; exceeding ±2 shows major asymmetry [21].

Available adjusted data of binary outcome were analyzed using the inverse variance method. The adjusted OR (aOR) was the effect size of choice for pooling of adjusted data. If an aOR was not provided, adjusted risk ratio or hazard ratio were converted to aOR. If the adjusted effect size with its 95% CI was not available and the intervention (daptomycin use) was not significantly associated with the outcome of interest, an adjusted effect size of 1 was imputed, and the standard error of the unadjusted analysis was utilized as the measure of dispersion, as described elsewhere [22].

Moreover, alongside the pooled effect sizes (of binary and continuous outcomes) and the 95% CIs thereof, a prediction interval was reported, reflecting the expected uncertainty in the summary effect if a new study was included in the meta-analysis [23].

Anticipated subgroup analyses concerned variables such as study design and place, different timing of mortality and relapse assessment, presence/absence of endocarditis, presence/absence of companion drugs and quality of the included studies.

For sensitivity analysis, how each individual study affected the overall estimate of the rest of the studies was evaluated by leave-one-out meta-analysis generation influential plots. Moreover, for quantitative bias analyses to assess the robustness of the results, the E-value was computed, that was defined as the minimum strength of association that an unmeasured confounder would need to have with both the treatment and the outcome to fully explain away a specific association; in this case, with the main outcome (mortality) [24].

Eventually, meta-regression analyses were planned to explore study-level sources of heterogeneity; for this purpose, when sample sizes, means and standard deviations were presented separately in each of the intervention groups as continuous moderators (for instance, age), the groups were combined in a single one in order to better assess the impact of a specific moderator on the dependent variable of interest. The Knapp–Hartung adjustment was implemented for random-effect model meta-regression [25].

All *p* values < 0.05 were considered statistically significant.

Analyses were performed through the following software: Review Manager (RevMan), version 5.4; MetaXL, version 5.3 (Ersatz, EpiGear International, Sunrise Beach, Australia); Comprehensive Meta-Analysis, version 3 (Biostat, Englewood, NJ, USA).

### 2.8. Ethics

This work relies on previously approved and conducted studies and is therefore exempt from ethics approval.

## 3. Results

### 3.1. Literature Search

The search strategy though three databases yielded 4394 records. A total of 2820 records were removed after de-duplication and a further 1550 were excluded by title/abstract screening. The full-text review focused on 24 papers, and 8 articles [26,27,28,29,30,31,32,33] were eventually included in the quantitative analysis. The entire screening and selection process is presented in Figure 1.

### 3.2. Study Description

The main features of the included studies are presented in Table 1. The majority were retrospective and observational in nature (seven out of eight), barring a post hoc subset analysis [26] of the seminal RCT on daptomycin use for Staphylococcus aureus bacteremia and right-sided endocarditis [11]. All studies were conducted in the United States, from 2001 to 2015, and involved adult subjects. In total, 1226 patients were included in the analysis, with the sample size ranging from 86 to 262 in the identified studies. In Table 1, further information is provided, such as dosages, administration as monotherapy or in combination, outcomes definitions and the corresponding figures expressed as proportions.

### 3.3. Quality of Included Studies

The details of quality assessment are described in Appendix A. Only the NOS tool, in a modified version, was utilized, since a study by Rehm and colleagues [26], being a post hoc analysis of randomized clinical trial addressing only a subset of patient, was considered as observational. Overall, the majority of the studies (seven out of nine) were deemed as low risk bias, while the remaining ones [27,28] were deemed as moderate risk due to the fact that they were not originally conceived for a pairwise comparison between daptomycin and vancomycin.

### 3.4. Mortality

All the included studies [26,27,28,29,30,31,32,33] compared mortality rate between patients undergoing DAP and the ones receiving VAN, although with different timing, which was the object of a subgroup analysis: 30-day, 60-day, all-cause/in-hospital. The overall death rate was 12.9% (72/554) in patients on daptomycin compared with 18.9% (127/672) in patients on VAN. From a meta-analytic standpoint, DAP use was associated with a protective effect on mortality, although this was not statistically significant: OR 0.73, 95% CI 0.40–1.33. Moderate heterogeneity was retrieved (I^2^ = 67%). No interaction was found between the subtotal estimates for the three identified subgroups, confirming the null hypothesis that homogeneity existed between the subgroup estimates of the population parameter (Figure 2). The 95% prediction interval was 0.12–4.42.

### 3.5. Clinical Failure

All included studies [26,27,28,29,30,31,32,33] evaluated the outcomes of the drugs under investigation in terms of clinical failure, usually a composite endpoint with heterogenous definitions across the various studies, as detailed in Table 1; failure encompassed mortality, relapse, microbiological/clinical cure, etc. DAP administration nearly halved the likelihood of clinical failure compared with vancomycin: OR 0.58, 95% CI 0.38–0.89 (I^2^ = 60%), as displayed in Figure 3. The 95% prediction interval was 0.16–2.04.

The meta-analysis of adjusted data confirmed a statistically significant difference in clinical failure between the two combination regimens (aOR against VAN use 1.94, 95% CI 1.33–2.82), as shown in Figure 4.

### 3.6. Infection Recurrence

Recurrence of infection was defined as relapse of infection within a pre-specified timeframe, 30 [28,29,30,31,32] or 42 days [26,27] after discharge or therapy completion. In the 30-day subgroup, DAP seemed to reduce the risk of relapse (OR 0.83, 95% CI 0.29–2.35, I^2^ = 42%); the contrary occurred in the 42-day subgroup (OR 1.36, 95% CI 0.53–3.44, I^2^ = 0%). Overall, no difference was observed (OR 1.05 95% CI 0.56–1.98, I^2^ = 10%), as displayed in Figure 5A. The 95% prediction interval was 0.38–2.94.

### 3.7. Infection Persistence

Persistence of bacteremia for at least 5 [31] or 7 days [30,32,33], also defined as ongoing positive cultures leading to discontinuation of the drug under investigation [26] and explored as secondary outcome by Usery et al. [28], was highlighted in 82 out of 437 patients on DAP (18.8%) and in 120 out of 479 patients on vancomycin (25.1%). Therefore, DAP was associated with an inferior risk of MRSA BSI persistence: OR 0.78, 95% CI 0.42–1.46, I^2^ = 69% (Figure 5B). The 95% prediction interval was 0.11–5.69.

### 3.8. Length of Stay

Data concerning LOS came from only four studies [28,30,32,33]. There was no sensible difference in mean about the duration of hospital stay between the two groups: mean difference −0.90 days, 95% CI −0.67 to 2.48; I^2^ = 16% (Figure 5C). The 95% prediction interval ranged from −0.94 to 1.16.

### 3.9. Safety Profile

In order to make the assessment of the safety profile of the two drugs more homogenous, only the rate of treatment-limiting AEs, the one prompting antibiotic discontinuation, was taken into account. Five studies provided evaluable data [26,29,30,32,33]: DAP was safer than VAN in a statistically significant fashion, with an OR of AEs equal to 0.15, 95% CI 0.06–0.36, I^2^ = 19% (Figure 5D). The 95% prediction interval was 0.03–0.94.

### 3.10. 30-Day Re-Admission

Re-admission risk after discharge was gauged by just three studies [27,30,32], within thirty days from discharge. No difference was detected between the two arms: OR 0.99, 95% CI 0.61–1.61, I^2^ = 0% (Figure 5E). The 95% prediction interval ranged from 0.33 to 2.77.

### 3.11. Sources of Heterogeneity and Sensitivity Analyses

To investigate sources of heterogeneity of results, subgroup analyses were performed. Some of them have already been presented (mortality and relapse according to the different reporting time) in the main analyses. All studies were conducted in the United States; all except one [26] were retrospective in nature and, among them, only one [30] was case–control by design, with the other ones being cohort studies. Due to these reasons, no subgroup analysis was carried out regarding study design and place.

With respect to study quality, endocarditis proportion, overall mortality rate and presence or absence of companion drugs, the results of subgroup analyses focused on the primary outcome are presented in Appendix A. The direction of effect was in favor of daptomycin, except when considering moderate-quality studies and studies in which the drugs under investigation were administered as monotherapy and were the same [27,28]: OR 2.45, 95% CI 1.11–5.40, and the test for subgroup difference was statistically significant. In a study by Rehm et al. [26], DAP was administered as monotherapy (instead VAN was combined with gentamycin), but its exclusion from the combination therapy subgroup did not affect the results, confirming the favorable effect on mortality of daptomycin compared to VAN.

To further address heterogeneity between studies as far as the main outcome was concerned, a meta-regression analysis based on covariates (when available) was performed, including (i) mean age (all studies), (ii) male proportion (7 studies), (iii) Apache score (4 studies), (iv) proportion of acute kidney injury at admission (4 studies), (v) proportion of infective endocarditis episodes (all studies), (vi) proportion of cases with minimal inhibitory concentrations (MICs) for vancomycin equal to 2 mg/L (5 studies) and (vii) proportion of cases treated with a concurrent agent (all studies, considering zero when only monotherapy was taken into account). Only the male proportion from seven studies [26,27,28,29,30,31,32] showed a significant, specifically negative, correlation with odds for mortality (Appendix A): slope coefficient −0.15%, 95%CI from −0.25 to −0.06, *p* = 0.009 (other data not shown). In other words, in studies showing more apparent benefit of DAP, the percentage of male patients was higher. This correlation was able to fully explain the between-study variance (adjusted R^2^ 100%).

The results of sensitivity analysis by means of leave-one-out method showed no relevant modification of the estimates after the exclusion of individual studies one by one for all outcomes (Appendix A).

Quantitative bias analysis demonstrated that results were not robust to unmeasured confounding: the E-value, estimating what the relative risk would have to be for any unmeasured confounder to overcome the observed association of DAP with mortality (the main outcome), was 1.62, being the lower 95% CI equal to 1.00 (Appendix A).

### 3.12. Publication Bias

The Doi plot for publication bias related to the main outcome showed major asymmetry, as confirmed by LFK index, which was equal to 2.64 (Appendix A).

### 3.13. Certainty of Evidence According to the GRADE Framework

A simplified version of a GRADE table assessing the certainty of evidence as to each outcome under investigation is presented as Appendix A. The nature of the studies informing the systematic review influenced the level of the evidence from the very beginning, and was set as low. Criteria to raise certainty to moderate were met only for the safety outcome (specifically, very large magnitude of effect, with OR < 0.2), whereas further research is likely to have an important impact on the confidence in the estimate of effect regarding the other endpoints.

## 4. Discussion

To the best of our knowledge, this is the first systematic review and meta-analysis that compares the efficacy and safety of daptomycin and vancomycin for the treatment of MRSA BSI and endocarditis.

Interestingly, the results of this study failed to show a statistically meaningful difference in terms of overall risk of mortality between the two arms of treatment, but a significant reduction in risk of clinical failure and treatment-limiting AEs was evidenced in favor of DAP. In addition, no differences were evidenced in terms of risk of 30-day readmission, infection persistence, relapse or length of stay.

These findings require careful interpretation. First, it should be considered that MRSA infections, particularly BSI and endocarditis, are archetypal “difficult-to-treat” diseases [34], posing a formidable clinical threat and imposing the evaluation of both antimicrobial therapy and adjunctive aspects of care, such as infectious disease consultation, echocardiography and source control in order to achieve success. Moreover, there is some consolidated literature regarding the importance of gain and loss of virulence determinants carried on by major genetic elements of MRSA [35] that play a vital role in bacterial adaptability, virulence [36] and patient survival [37].

Second, beyond the pathogen itself, the extreme variability in clinical settings (intensive care units, post-surgical, general wards, etc.), baseline patient characteristics (immunocompromised vs. immunocompetent, sepsis/septic shock vs. sub-acute infections, old vs. young, etc.) and type of infection (primary BSI, device-associated BSI, endocarditis, presence of secondary abscesses) significantly impacts the overall risk of mortality, 30-day readmission and risk of infection relapse; indeed, outcomes after MRSA BSI are generally worse than those after MSSA infections thanks to the former’s confounding associations with other prognostic factors [38].

Third, even after adjusting for all confounders, the management of MRSA BSI still requires further insight. Another factor to be taken into account is the duration of S. aureus bacteriemia, as this is an important and independent predictor of poor outcomes [39]. Therefore, future studies on *S. aureus* BSI should consider time to bacteriemia clearance as a pivotal variable.

Of note, sensitivity analyses according to the leave-one-out method did not affect the results regarding mortality, confirming a slightly favorable, but not significant, positive effect of DAP on mortality, although the prediction interval included the opposite effect. On the other hand, results from subgroup analysis according to study quality differed diametrically; a statistically and clinically meaningful effect of daptomycin on mortality (OR 0.51, 95% CI 0.30–0.86) emerged when considering only studies at low risk of bias, whereas totally opposite results stemmed from pooling data of studies at moderate risk of bias (OR 2.45, 95% CI 1.11–5.40) [27,28]. A significant interaction existed between the subtotal estimates for the two subgroups (*p* = 0.001), hinting at the estimation of different population parameters. As matter of fact, these two studies were downgraded in the risk of bias assessment since they were not devised to perform a head-to-head comparison between daptomycin; one was focused on patients receiving ceftaroline, matched with subjects undergoing VAN or DAP [27], while another one was a simple retrospective study evaluating different regimens against MRSA BSI, specifically linezolid, as well as VAN or DAP [28]. Interestingly, the abovementioned subgroup overlapped with the one linked with the presence or absence of companion drugs. Combination therapy was present in six studies and was described in a fraction of patients ranging from 2% [31] to 86% [32]; a beta-lactam agent, an aminoglycoside or rifampin were the concurrent drugs of either DAP or VAN. Therefore, the direction of the effect was completely different when considering studies allowing combination regimens (DAP halving mortality rate) separated from studies relying on monotherapy (DAP associated with a more than two-fold mortality risk). However, when excluding the only prospective study [26] in which DAP was administered alone from the first subgroup, while VAN was administered along with gentamycin, the protective effect of daptomycin on death risk was even higher (OR 0.44, 95% CI 0.29–0.66). The role of combination therapy for MRSA BSI is still being debated. A recent meta-analysis of 15 studies (3 RCTs) on 2594 patients investigating the role of adjuvant beta-lactam therapy failed to demonstrate an overall benefit on mortality of combination regimens compared with DAP or VAN alone (relative risk 1.14. 95% CI 0.92–1.57), although a positive effect was apparent regarding clinical failure, recurrence or persistence of bacteremia, as well as on death rate when considering the small subgroup (three studies) of patients receiving DAP plus beta-lactam [40]. The association between DAP and a beta-lactam agent is intriguing since it might help to circumvent the emergence of strains non-susceptible to DAP and the emergence of resistance to host cationic antimicrobial peptides; by a so-called “see-saw” effect mechanism, MRSA strains might gain improved susceptibility to beta-lactams [41]. At any rate, no definitive evidence can be drawn to date on the superiority of one approach over the other in the context of MRSA BSI, though current guidelines back monotherapy [42,43]; however, the addition of another agent can be taken into account when endocarditis is present as a high-inoculum infection [42,44].

Data could not be split to focus on infective endocarditis (IE) cases separately, but the pooled effect size did not change when considering studies with higher (at least 20%) or lower proportions of IE cases.

Several univariate meta-regression analyses were run to further explore heterogeneity with regard to the main outcome. The only factor significantly associated with the effect size and able to fully explain the between-study variance was the male proportion. This finding is not simple to interpret. Moreover, a gender effect has been supposed for MRSA BSI; specifically, female subjects seem to show a poorer prognosis [45,46]. The better outcome for males associated with DAP use needs to be further elucidated. Meta-regression of patients’ baseline features such as age, severity (Apache score) and renal function did not yield relevant results. The same outcome was applicable to important microbiological predictors, such as the number of cases with MICs for VAN equal to 2 mg/L, that could not be analyzed separately as distinct subgroup. A recent meta-analysis [47] confirmed seminal results from Kalil and colleagues [48] in the context of MRSA BSI; high VAN MICs in the susceptible range of MRSA isolates (≥1.5 mg/L) did not augment the likelihood of overall mortality, though the possible confounding effect of more frequent prescription of combination antimicrobial therapy in this subgroup should not be overlooked.

While the comparison between DAP and VAN did not yield statistically significant results regarding mortality, the scenario was different regarding clinical failure, the most important secondary outcome. DAP was associated with a meaningful reduction in the risk of clinical failure (OR 0.58, 95% CI 0.38–0.89), in spite of the contrasting results from some studies [27,28]. That held true when performing an analysis of adjusted effect estimates, carried out according to the current recommendation for the reporting of evidence synthesis of observational studies [49], confirming that VAN nearly doubled the probability of clinical failure. However, this endpoint was described inhomogeneously across the studies, often encompassing a composite outcome including mortality and other types of events (such as recurrence, microbiological persistence, etc.). Of course, infection resolution cannot prevent each fatal event, especially in a framework of non-attributable mortality, and that may explain the different results between the main outcome and secondary ones. Greater chances of a cure from infection linked with DAP might lie in the higher bactericidal activity of daptomycin over vancomycin, especially in cases of deep-seated infections, giving higher chances to achieve a quick bacterial clearance and reducing the risk of developing further metastatic foci [50].

The largest effect was retrieved when investigating the rate of treatment discontinuation due to AEs, which was notably lower in the DAP group (OR 0.15, 95% CI 0.06–0.36). This was the only in case in which the 95% prediction interval did not include the null effect or the opposite effect (0.03–0.94). However, only five studies [26,29,30,33] presented useful data for meta-analytical purposes. The safety profile of VAN is a well-known issue; the antibiotic should be administered through a central line [51] and, most importantly, should undergo strict therapeutic drug monitoring (TDM) [52]. For many years, TDM has essentially been based on trough-only sampling, but recent guidelines advocate an area under the curve (AUC) over the 24 h/MIC ratio of ≥400, as long as the MRSA strain shows an MIC lower than 1 mg/L as the paramount pharmacokinetics/pharmacodynamic predictor of VAN activity in order to maximize its clinical efficacy while minimizing its toxicity [53]. The use of AUC/MIC ratio for VAN is still controversial [54], and outcomes data for an MIC of 2 mg/L are quite limited [53].

As far as the other secondary outcomes were concerned (recurrence, persistence of infection, LOS and 30-day readmission), no significant differences were detected among the two drugs.

Globally, the results of the present work seem to point in the direction of DAP as the preferred agent for MRSA BSI/endocarditis, rather than simply as an alternative to VAN; this is reiterated by recent consensus [43]. Moreover, the introduction of the generic version of DAP in 2016 should make use of drug more appealing for both prescribers and C-suite, although pharmacoeconomic analyses conducted more than a decade ago demonstrated that the cost-effectiveness of DAP and VAN was similar [55]. Results of our evidence synthesis are in line with recent works on this topic. For instance, Schweizer et al. showed that, by shifting to daptomycin, patients previously treated with vancomycin and who still presented persistent bacteriemia after 3 days of therapy significantly reduced the risk of mortality, indicating a possible theorical superiority of one drug over the other for this specific setting [56].

The strengths of our study include protocol pre-registration, a thorough literature search, strict inclusion criteria, multiple subgroups and sensitivity as well as meta-regression analyses, not to mention our quantitative bias analysis. Nevertheless, there are also several limitations to this work. These comprise the observational and retrospective nature of the majority of the included studies, their limited number, relatively heterogeneous populations and inconsistencies in relation to clinical failure definitions. Small-study effect could not be excluded, as highlighted by the LFK index. The results regarding the main outcome were not robust to unmeasured confounding, since the calculated E-value was 1.62, meaning that residual confounding could explain away the observed association as to whether an unmeasured covariate existed with a relative risk association at least as large as 1.62 for both DAP use and mortality. The prediction intervals included the opposite effect in each case, except for treatment discontinuation; therefore, in future patients, an effect in favor of VAN cannot be ruled out. A proper comparison in the subgroup of IE could not be performed, and the impact of different dosing or companion agents could not be properly accounted for. Indeed, it is well known that doses higher than the DAP label dose (7 mg/kg or greater) are associated with better outcomes in the context of MRSA BSI [57]. Finally, the impact of source control (when performed) and different origins of BSI could not be assessed due to the paucity of data across the study.

## 5. Conclusions

The results of our meta-analysis, the largest and the first that has been completed on this topic to date, suggest that daptomycin use is associated with a reduced clinical failure and is better tolerated than vancomycin for the treatment of patients with MRSA infections without lung involvement. Conversely, no significant benefit in all-cause mortality, risk of infection persistence, 30-day re-admission and infection relapse were noticed, although these aspects were probably influenced by other factors (presence of devices, deep-seated infections, device removal or surgical procedures in cases of endocarditis). Notably, these findings were mostly derived from retrospective and observational studies, which weakens the findings of this study.

Nevertheless, due to its confirmed better tolerability and bactericidal activity, daptomycin might be considered as the treatment of choice for MRSA BSI, also considering its favorable pharmacokinetic profile that allows a once daily administration without the requirement for systematic therapeutic drug monitoring to achieve efficacy and preserve tolerability.

Further research, particularly represented by RCTs, is needed to evaluate patient-specific factors to guide antibiotic selection and other treatment strategies in patients with MRSA BSI and endocarditis.

## Figures and Tables

**Figure 1 antibiotics-10-01014-f001:**
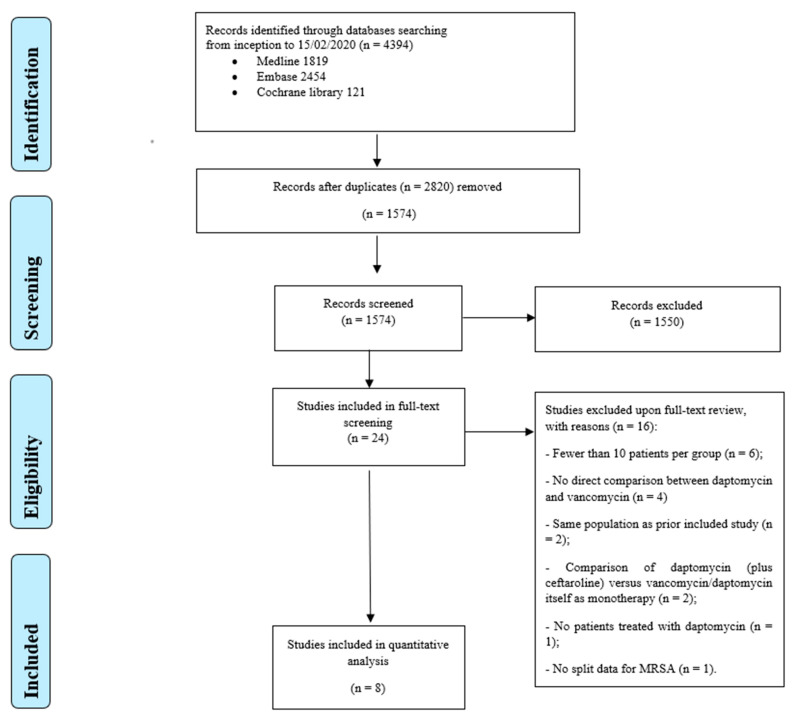
Results of literature search and flow diagram for selection of eligible studies.

**Figure 2 antibiotics-10-01014-f002:**
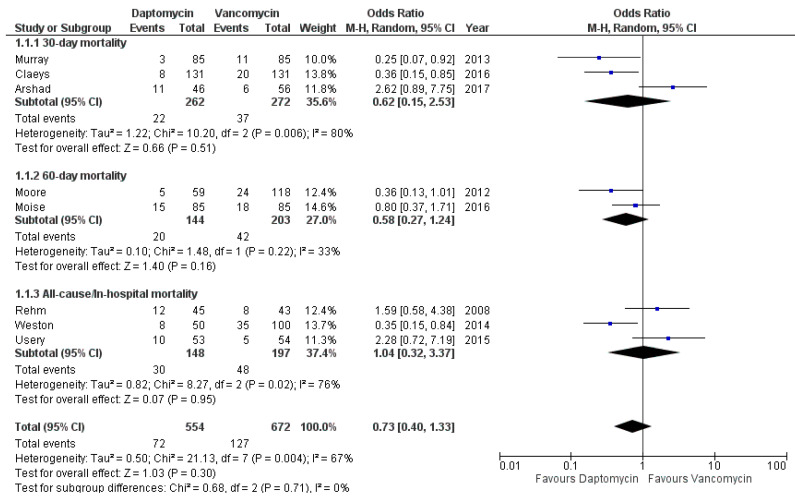
Forest plot depicting the odds ratios of mortality of patients receiving daptomycin vs. vancomycin.

**Figure 3 antibiotics-10-01014-f003:**
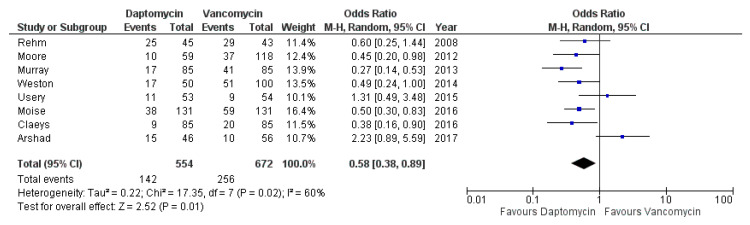
Forest plot depicting the odds ratio of clinical failure of patients receiving daptomycin vs. vancomycin.

**Figure 4 antibiotics-10-01014-f004:**
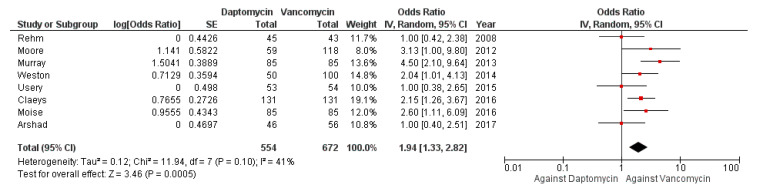
Forest plot depicting the adjusted odds ratio of clinical failure of patients receiving daptomycin vs. vancomycin.

**Figure 5 antibiotics-10-01014-f005:**
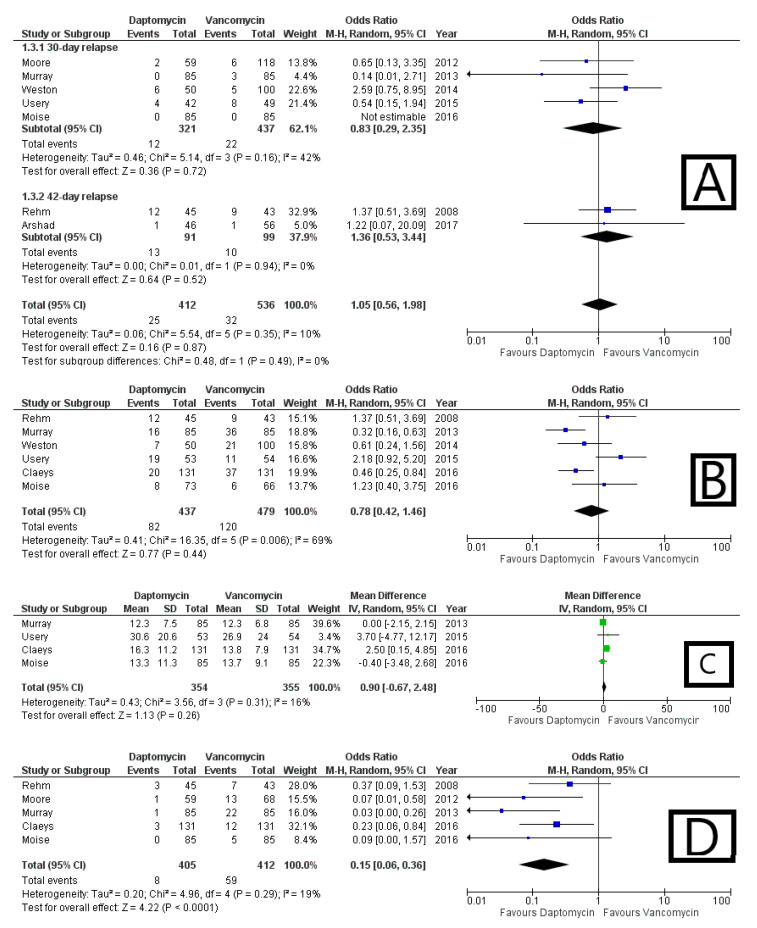
Forest plot depicting the odds ratio of recurrence (**A**) and BSI persistence (**B**), the mean difference in LOS (**C**), as well as the odds ratio of discontinuation due to safety issues (**D**) and 30-day readmission (**E**) of patients receiving daptomycin vs. vancomycin.

**Table 1 antibiotics-10-01014-t001:** Overview of included studies’ main features.

Author, yr [Reference]	Country	Design	No. of Centers	Study Period	Sample Size	MRSA Associated Endocarditis	Group		Primary Endpoints ^1^	Mortality	Clinical Failure (Composite Outcome)	Relapse	Persistent BSI	Safety Assessment ^2^
							Daptomycin (Daily Dose, Treatment Duration, Combination Therapy [%, Drug])	Vancomycin (Trough Concentration or Daily Dose, Treatment Duration, Combination Therapy [%, Drug])						
**Arshad S et al. 2017** [27]	USA	Retrospective matched cohort study	1	Nov 2009–Dec 2013	102 (46 DAP vs. 56 VAN)	DAP: 19/46 VAN: 13/56	N/A 4–8 weeks N/A	N/A 4–8 weeks N/A	1–3, 5	DAP: 11/46 VAN: 6/56	DAP: 15/46 10/56	DAP: 1/46 1/56	N/A	N/A
**Claeys C et al. 2016** [33]	USA	Retrospective matched cohort study	3	Jan 2010–Mar 2015	262 (131 VAN vs. 131 DAP)	DAP: 25/131 VAN: 25/131	8.2 mg/kg (IQR, 6.4–10.0) N/A 24.4% (13% ceftaroline, 6.9% rifampin)	17.7 mg/L (IQR, 13–222.0) N/A 20.6% (10.7% ceftaroline, 4.6% rifampin)	1–6	DAP: 8/131 VAN: 20/131	DAP: 38/131 VAN: 59/131	DAP: 0/131 VAN: 0/131	DAP: 20/131 VAN: 37/131	DAP: 2 (3/131) VAN: 1 (12/131)
**Moise PA et al. 2016** [32]	USA	Retrospective matched cohort study	11	2005–2012	170 (85 DAP vs. 85 VAN)	31/170 (no group subdivision)	6 mg/kg (IQR 6–8) 16 days (IQR, 10–35) 74% (13% gentamicin, 15% rifampin, 46% β-lactam)	17.5 mg/L (IQR,14.0–22.0) 16 days (IQR, 9–31) 98% (16% gentamicin, 11% rifampin, 71% β-lactam)	1–4, 6	DAP: 15/85 VAN: 18/85	DAP: 9/85 VAN: 20/85	DAP: 0/85 VAN: 0/85	DAP: 8/72 VAN: 6/66	DAP: 1 (6/64) VAN: 1 (14/62)
**Moore CL et al. 2012** [29]	USA	Retrospective case–control study	1	2005–2009	177 (59 DAP vs. 118 VAN)	DAP: 17/59 VAN: 34/118	6 mg/kg (dose adjustment if GFR ≤30 mL/min) 12 days (IQR, 8–18) 38% (aminoglycoside or rifampin for ≥3 days)	10-20 mg/L 15 days (IQR, 10–24) 51% (aminoglycoside or rifampin for ≥3 days)	1–4, 6	DAP: 5/59 VAN: 24/118	DAP: 10/59 VAN: 37/118	DAP: 2/59 VAN: 6/118	N/A	DAP: 1 (1/59) VAN: 1 (13/63)
**Weston A et al. 2014** [31]	USA	Retrospective case–control study	1	Jan 2001–Aug 2011	267 (50 DAP vs. 100 VAN)	DAP: 13/50 VAN: 11/100	6.8 mg/kg (range, 5.1–10.8) (dose adjustment if GFR ≤30 mL/min) Average 28 days N/A	15.3 μg/mL (range, 8.2–25.6) Average 21 days N/A	1–4	DAP: 8/50 VAN: 35/100	DAP: 17/50 VAN: 51/100	DAP: 6/50 VAN: 5/100	DAP: 7/50 VAN: 21/100	DAP: 2 (2/50) VAN: N/A
**Murray KP et al. 2013** [30]	USA	Retrospective matched cohort study	4	Jan 2005–Mar 2012	170 (85 DAP vs. 85 VAN)	DAP: 20/85 VAN: 20/85	8.4 mg/kg (IQR, 6.3–9.9) 10 days (IQR, 8–17) 30.6% (14.1% aminoglycoside, 16.5% rifampin)	17.6 µg/mL (IQR, 14.9–21.2) 9 days (IQR, 6-16) 47.1% (25.9% aminoglycoside, 21.2% rifampin)	1–4	DAP: 3/85 VAN: 11/85	DAP: 17/85 VAN: 41/85	DAP: 0/85 VAN: 3/85	DAP: 16/85 VAN: 36/85	DAP: 2 (1/85) VAN: 1 (22/85)
**Usery JB et al. 2015** [28]	USA	Retrospective cohort study	1	Jun 2008–Nov 2010	122 (53 DAP vs. 54 VAN)	DAP: 6/53 VAN: 6/54	6.7 mg/kg (range, 4.9–8.5) 16.4 days (range, 6.8–26) N/A	13.6 mg/kg (range, 9.6–17.6) 13.6 days (range, 6.5–20.7) N/A	1–3, 7	DAP: 10/53 VAN: 5/54	DAP: 11/53 VAN: 9/54	DAP: 4/42 VAN: 8/49	DAP: 19/53 VAN: 11/54	N/A
**Rehm SJ et al. 2008** [26]	International	Subset analysis of an open-label randomized trial	Multicentre	Aug 2002–Mar 2005	88 (45 DAP vs. 43 VAN plus gentamicin)	DAP: 13/45 VAN: 10/43	6 mg/kg 10-14 days for uncomplicated bacteraemia; at least 28 days for complicated bacteraemia and endocarditis N/A	14.9 μg/mL 10-14 days for uncomplicated bacteraemia; at least 28 days for complicated bacteraemia and endocarditis 100% (gentamicin 1 mg/kg every 8 hours for the first 4 days of treatment)	1–4, 6, 7	DAP: 12/45 VAN: 8/43	DAP: 25/45 VAN: 29/43	DAP: 12/45 VAN: 9/43	DAP: 3/45 VAN: 7/43	DAP: 1, 3 (3/45) VAN: 1, 3 (7/43)

^1^ Endpoints include the following: (1) mortality (including all-cause mortality, in-hospital mortality, 30-day mortality, 60-day mortality); (2) clinical failure (composite endpoint); (3) relapse (including 30/42-day relapse, 60-day MRSA BSI-related re-admission); (4) persistent BSI (including bacteraemia persistent ≥5/7 days, microbiologic failure); (5) duration of hospitalization and therapy; (6) safety; (7) success (including clinical cure, negative blood cultures after 42 days since end of therapy). ^2^ Safety assessment includes the following adverse events: (1) nephrotoxicity; (2) CK elevation; (3) all AEs.

## Data Availability

Dataset is available from the corresponding author after reasonable request.

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
