# Peer review of "Daptomycin versus Vancomycin for the Treatment of Methicillin-Resistant Staphylococcus aureus Bloodstream Infection with or without Endocarditis: A Systematic Review and Meta-Analysis"

_antibiotics, 2021, doi:10.3390/antibiotics10081014_

Round 1

Reviewer 1 Report

The research topic is important in clinical practice because BSI of MRSA is a fatal infection and complications can be a problem for patients. Therefore, I think it is a suitable research topic.

  1. The papers covered in this study are up to 2015 and do not include the recent studies. Several papers on the subject were published after the period covered by this study. Some RCTs have been conducted in studies after the period covered by this study, and they have examined the same issues as this study. At least those studies should be included. In conclusion, no new findings were obtained in this study.

  1. Other studies have indicated that daptomycin is tolerated for long term administration, and other studies have indicated that vancomin is suitable for patients with pulmonary involvement. The final conclusions, such as mortality, are also not significantly different between the two drugs. So, nothing from this paper will affect the actual choice of MRSA BSI treatment.

  1. The fact that there is no significant difference in outcomes between the two drugs is an important message, but that does not change the existing studies.

Author Response

Reviewer 1

The research topic is important in clinical practice because BSI of MRSA is a fatal infection and complications can be a problem for patients. Therefore, I think it is a suitable research topic.

R1: First of all, thanks for the attention you paid to our manuscript and your favorable comments.

The papers covered in this study are up to 2015 and do not include the recent studies. Several papers on the subject were published after the period covered by this study. Some RCTs have been conducted in studies after the period covered by this study, and they have examined the same issues as this study. At least those studies should be included. In conclusion, no new findings were obtained in this study.

R1: Unfortunately the latest included study dates back to 2015, but more recent studies specifically comparing daptomycin versus vancomycin are lacking. Some studies investigating the role of daptomycin against MRSA BSI, especially in the contest of monotherapy vs combination therapy are available, but no further comparative studies between vancomycin and daptomycin are available.

Other studies have indicated that daptomycin is tolerated for long term administration, and other studies have indicated that vancomin is suitable for patients with pulmonary involvement. The final conclusions, such as mortality, are also not significantly different between the two drugs. So, nothing from this paper will affect the actual choice of MRSA BSI treatment.

R2: We fully acknowledge that our results will not profoundly change the management of BSI, but it is the first systematic review that summarizes all the available evidence regarding the comparison between daptomycin and vancomycin, and might serve as basis for further studies as RCTs.

The fact that there is no significant difference in outcomes between the two drugs is an important message, but that does not change the existing studies.

R3: Please see the previous reply.

Reviewer 2 Report

This paper is a systematic review and meta-analysis evaluating the efficacy of daptomycin for the treatment of MRSA BSI with or without endocarditis compared with vancomycin. The paper concluded daptomycin can reduce clinical failure and have better tolerability compared to vancomycin, although it was not beneficial for reducing mortality and other clinical outcomes.

These findings are valuable but there are some concerns in this paper as below:

  1. The main purpose of this meta-analysis was to evaluate the statistical superiority in all-cause mortality in daptomycin for the treatment of MRSA BSI compared with vancomycin and the authors failed to demonstrate the superiority, even though the authors could find the superiority of daptomycin in two secondary-outcomes. The authors should describe these facts in Abstract and conclusion, sincerely.
  2. As the authors described in Discussion, most of included studies in this meta-analysis were retrospective and observational studies at a single institute and there were no randomized control studies, and there has several vias such as publication bias. The reviewer cannot deny that these limitations reduced the value of paper, even though the results of this study were valuable and further meta-analyses with several RCT with larger cohorts will be needed to conclude the superiority of daptomycin compared to vancomycin in the future. Please describe the limitation of this study and the necessity of further meta-analyses more clarify and detailly.
  3. The author described “A total of 2830 records were removed after de-duplication and further 1548 were excluded by title/abstract screening. The full-text review focused on 24 papers, and …” in page 8 line 186-188 and Figure1. In contrast, 1574 minus 1548 was 26 but not 24. Please confirm the number in main text and Figure1.
  4. There were lacks of the spaces between words in many points in the main text. Please confirm the whole manuscript and insert spaces appropriately.

Author Response

Reviewer 2

This paper is a systematic review and meta-analysis evaluating the efficacy of daptomycin for the treatment of MRSA BSI with or without endocarditis compared with vancomycin. The paper concluded daptomycin can reduce clinical failure and have better tolerability compared to vancomycin, although it was not beneficial for reducing mortality and other clinical outcomes.

These findings are valuable but there are some concerns in this paper as below:

The main purpose of this meta-analysis was to evaluate the statistical superiority in all-cause mortality in daptomycin for the treatment of MRSA BSI compared with vancomycin and the authors failed to demonstrate the superiority, even though the authors could find the superiority of daptomycin in two secondary-outcomes. The authors should describe these facts in Abstract and conclusion, sincerely.

R1: First of all, thanks for the attention you paid to our manuscript. Actually, the purpose of our work was to assess in a meta-analytic way potential difference between DAP and VAN but no statistically superiority of either was assumed, since, differently from a randomized critical trial, this kind of assumption is not required. We feel that the abstract and the conclusion correctly reflect the results of the analyses.

As the authors described in Discussion, most of included studies in this meta-analysis were retrospective and observational studies at a single institute and there were no randomized control studies, and there has several vias such as publication bias. The reviewer cannot deny that these limitations reduced the value of paper, even though the results of this study were valuable and further meta-analyses with several RCT with larger cohorts will be needed to conclude the superiority of daptomycin compared to vancomycin in the future. Please describe the limitation of this study and the necessity of further meta-analyses more clarify and detailly.

R2: The discussion presents a long section describing all the limitations of the paper as you suggest. At any rate, the necessity of well-conducted RCTs is reinforced in the ending part of the discussion.

The author described “A total of 2830 records were removed after de-duplication and further 1548 were excluded by title/abstract screening. The full-text review focused on 24 papers, and …” in page 8 line 186-188 and Figure1. In contrast, 1574 minus 1548 was 26 but not 24. Please confirm the number in main text and Figure1.

R3: Thanks for your careful review. Both the figures and the text have been amended, since the number of excluded records was 1550.

There were lacks of the spaces between words in many points in the main text. Please confirm the whole manuscript and insert spaces appropriately.

R4: There was some kind of problem when uploading the file but the revised manuscript has been prepared according to the editorial rules, by using the appropriate template, being careful to correct lack of spaces or other orthographic issues.

Round 2

Reviewer 1 Report

Daptomycin was originally approved for the treatment of MRSA infections due to its non-inferiority compared to vancomycin. This study does not present any new findings.